# Large-Area Film Thickness Identification of Transparent Glass by Hyperspectral Imaging

**DOI:** 10.3390/s24165094

**Published:** 2024-08-06

**Authors:** Shuan-Yu Huang, Riya Karmakar, Yu-Yang Chen, Wei-Chin Hung, Arvind Mukundan, Hsiang-Chen Wang

**Affiliations:** 1Department of Optometry, Central Taiwan University of Science and Technology, Taichung 40601, Taiwan; 108695@ctust.edu.tw; 2Department of Mechanical Engineering, National Chung Cheng University, Chiayi 62102, Taiwan; karmakarriya345@gmail.com (R.K.); a15937a5566@gmail.com (Y.-Y.C.); 3Department of Physics, R. O. C. Military Academy, Kaohsiung 83020, Taiwan; hung.wc0602@msa.hinet.net; 4Hitspectra Intelligent Technology Co., Ltd., Kaohsiung 80661, Taiwan

**Keywords:** hyperspectral imaging, near-infrared, transparent glass, thickness estimation

## Abstract

This study introduces a novel method for detecting and measuring transparent glass sheets using hyperspectral imaging (HSI). The main goal of this study is to create a conversion technique that can accurately display spectral information from collected images, particularly in the visible light spectrum (VIS) and near-infrared (NIR) areas. This technique enables the capture of relevant spectral data when used with images provided by industrial cameras. The next step in this investigation is using principal component analysis to examine the obtained hyperspectral images derived from different treated glass samples. This analytical procedure standardizes the magnitude of light wavelengths that are inherent in the HSI images. The simulated spectral profiles are obtained using the generalized inverse matrix technique on the normalized HSI images. These profiles are then matched with spectroscopic data obtained from microscopic imaging, resulting in the observation of distinct dispersion patterns. The novel use of images coloring methods effectively displays the thickness of the glass processing sheet in a visually noticeable way. Based on empirical research, changes in the thickness of the glass coating in the NIR-HSI range cause significant changes in the transmission of infrared light at different wavelengths within the NIR spectrum. This phenomenon serves as the foundation for the study of film thickness. The root mean square error inside the NIR area is impressively low, calculated to be just 0.02. This highlights the high level of accuracy achieved by the technique stated above. Potential areas of investigation that arise from this study are incorporating the proposed approach into the design of a real-time, wide-scale automated optical inspection system.

## 1. Introduction

The wide-ranging uses of processed glass products, including household glass, touchscreen panels for mobile electronic devices [1], construction materials [2], industrial strengthened glass, coating material [3], and automotive glass [4], have led to the need for the development of different coating properties to fulfill various purposes [5,6,7]. The thickness of these coatings varies, and they possess certain capabilities, including anti-oxidation, corrosion resistance, high-temperature tolerance, and ease of cleaning [8,9,10]. Glass, as a substance, has a vast range of compositions, leading to the existence of several unique forms of glass globally [11,12]. The present study examines various types of energy-efficient glass that are transparent. These include soda-lime glass (also known as calcium sodium glass), white glass (such as Germany’s Schott B270 and D263T), quartz glass, specialty strengthened glass (like Corning Gorilla), crown glass (such as K9/BK7), borosilicate float glass (like Schott BOROFLOAT 33), and sapphire glass. These types of glass have been extensively used and studied in various applications [13,14,15]. Sapphire glass is used in LED manufacture, microelectronic circuits, and as protective optical glasses [16,17]. Glass functionalities can be categorized into different types, such as heat-absorbing glass that has body coloring and surface coatings, heat-reflective glass that has thin films to reflect solar energy (with a reflectance of 20–40%), low-emissivity glass that has coatings to reflect far-infrared radiation (reducing radiative heat transfer by more than 30%), and insulated glass that consists of multiple glass layers separated by dry gas (reducing thermal conductivity by 40%).

Unprocessed glass usually has a high rate of transmitting solar and ultraviolet (UV) rays, surpassing 40%. This may result in a considerable increase in interior temperature [18]. Energy-efficient glass may effectively balance illumination levels while minimizing solar and UV transmission rates to less than 30%. The current techniques for applying coatings to glass mainly include two methods: vacuum magnetron sputtering and chemical vapor deposition (CVD) [19,20,21,22]. Vacuum magnetron sputtering is a process that includes depositing coating materials onto glass surfaces in a layer-by-layer manner utilizing magnetron sputtering technology [23]. This technique yields a diverse range of hues on a transparent glass base. Due to its exceptional corrosion and abrasion resistant qualities, it is commonly used as a manufacturing process. CVD is a process where reaction gases are introduced into a manufacturing line and then decomposed on a heated glass surface. This leads to the uniform deposition of the coating material [24]. The benefits of this technology include little investment in equipment, effortless management, cheap costs of manufacturing, and exceptional chemical stability [25]. Moreover, it facilitates heat processing, making it a very auspicious manufacturing technique. The coatings used for energy-efficient glass mainly consist of polyurethane (PU) multifunctional polymers, UV absorbers (Picasus UV), and nano-transparent heat-insulating composite coatings. Ensuring product quality control during manufacture of glass coatings is challenging due to their clear and colorless nature [26]. The task of maintaining constant transparency and quality in clear, colorless glass coatings presents difficulties with homogeneity, contaminant control, and adherence to the underlying surface. It is crucial to maintain consistent thickness and surface cleanliness, while also maintaining strong adhesion without any deformations. To achieve the strict requirements for these coatings, it is crucial to implement rigorous quality control measures, accurate curing procedures, and constant monitoring of environmental variables. Hence, the glass sector requires a highly proficient and productive inspection system.

Groot et al. developed a white-light interferometer that utilizes novel signal processing methods to obtain the 3D top surface and thickness profiles of transparent films [27]. Nevertheless, this technique is restricted to transparent films that fall within the thickness range of 10 nm to 10 µm. Taherimakhsousi et al. provide a flexible convolutional neural network (CNN) that can effectively analyze thin-film images. This CNN is capable of detecting and measuring various faults, and it can be used in different materials and imaging scenarios [28]. Nevertheless, the investigation was carried out only on a restricted number of samples. The study conducted by Beliaev et al. examines the optical, structural, and compositional characteristics of silicon nitride films that were formed using reactive radio-frequency low-pressure and plasma-enhanced CVD [29]. However, the research does not directly relate to the identification and measurement of clear glass films. The majority of existing systems use conventional image processing techniques that just utilize the three primary color channels (red, green, and blue). One of the advantages of an HSI system is its ability to process images across a broad electromagnetic spectrum (ranging from 250 nm to 15,000 nm, including thermal infrared), as opposed to traditional image processing techniques that only utilize the three primary colors (red, green, and blue, or RGB) [30,31,32]. However, the potential benefits of HSI have yet to be completely harnessed in the automated optical inspection sector for coated glasses. However, ambient illumination and surface reflections can dramatically impact color-based systems. These factors can cause color measuring errors, making results unreliable. This constraint is especially problematic in industrial settings where controlled illumination is not always possible.

Therefore, we introduce an innovative approach for detecting and measuring transparent glass sheets. This technique utilizes HSI technology, enabling a digital camera to generate images in both the VIS and NIR spectral bands. The approach classifies the changes in thickness of the glass films into seven unique groups by examining the differences in spectral characteristics across these bands. The remaining portions of this investigation are organized into four clearly defined sections. Section 2 provides a comprehensive explanation of the algorithms developed for VIS-HSI and NIR-HSI within the scope of this research undertaking. The findings of this investigation are presented in Section 3. Section 4 and Section 5 of this research provide an overview of the discussion and conclusion of this study, respectively.

## 2. Materials and Methods

The present study used the vacuum sputtering method to deposit various film thicknesses (4000 nm, 5000 nm, 6000 nm, 7000 nm, 12,000 nm, and 14,000 nm) onto Low-E glass. The specimens used in this study were labeled as G4000, G5000, G6000, G7000, G12000, and G14000, according to their respective thicknesses, as seen in Figure 1a–f. Figure 2 shows the whole process of this study. In this experimental study, six clear film glass samples with varying thickness values were subjected to measurement. To verify the thickness of the glass film, the first procedure involves using a surface profilometer (KLA-Tencor D300 Profiler, KLA, Zhubei City, Taiwan, Appendix A) to assess the surface profile and ascertain the precision of the glass film’s thickness. In the subsequent stage, A dual-beam spectrophotometer (CT-8600, Chrom Tech, Taipei, Taiwan, Appendix A) is used to quantify the transmittance of the VIS and infrared (IR) spectral bands, therefore acquiring the distinctive properties of each glass film in terms of their VIS and NIR transmittance. These processes can build up the gold sample database for the connection between the thickness and spectrum of the samples. The different thickness samples can generate principal component analysis (PCA) regions. These regions can be used as a basis for judging the thickness of the samples. The third phase involves the use of VIS-HSI (visible hyperspectral imaging, 380–780 nm) and NIR-HSI (near-infrared hyperspectral imaging, 780–1100 nm) technologies to acquire simulated spectral data spanning from VIS to NIR bands. Finally, the NIR spectral data are employed to quantify the thickness of the transparent glass layer, therefore acquiring the spectrum intensity of NIR light transmittance. A glass film detecting system is developed using custom color planning based on the signal intensity of NIR spectral transmittance. The data acquisition system integrates an RGB camera, a SWIR camera, and spectrometers. The RGB and SWIR cameras capture spatial information across distinct wavelength ranges, while the spectrometers yield accurate spectral data. This integration allows us to restore HSI data with a combination of excellent spatial and spectral resolution. This approach utilizes the advantages of both camera systems and spectrometers to accomplish thorough hyperspectral imaging (HSI) data collection.

### 2.1. Visible Hyperspectral Imaging Algorithm

The present study utilizes VIS-HSI, which is achieved through the integration of an industrial camera (model DFK 33UX265; The Imaging Source: New Taipei City, Taiwan) and a visible hyperspectral algorithm (VIS-HSI). Operating within the wavelength range of 380 nm to 780 nm, with a spectral resolution of 1 nm, this process necessitates the determination of a conversion matrix that bridges the industrial camera and the spectrometer (QE65000, Ocean Optics: Dunedin, FL, USA), underpinning the establishment of the VIS-HSI methodology (see Appendix A for the schematics of the VIS-HSI algorithm). The fundamental premise that underlies VIS-HSI involves the derivation of the aforementioned conversion matrix, which is pivotal in affecting the transmutation of captured digital imagery into hyperspectral format. The calibration of this transformation is realized through meticulous analysis that employs multiple common reference targets. For the purposes of this study, the 24-color card (X-Rite Classic, 24 Color Checker; X-Rite: Grand Rapids, MI, USA) was selected as the reference object. This choice was based on the inclusion of pivotal hues, such as blue, green, red, and gray, along with a comprehensive representation of prevalent natural colors.

The camera system is prone to variations, such as white balance, which might lead to possible inaccuracies that need to be corrected before further processing. To address this problem, a recommended procedure involves incorporating a uniform 24-color card into the camera–spectrometer setup. This will allow for the capture of images of the 24-color patches using the sRGB color space with an 8-bit depth. Additionally, it will enable the collection of reflectance spectrum data for the 24-color blocks, which cover the entire spectral range from 380 nm to 780 nm. The data are thereafter converted into the CIE 1931 XYZ color space, as described in Appendix A, which provides the specific equations utilized in this research. In the field of image processing, the images that are acquired (saved in JPEG format, with a depth of 8 bits) are first placed in the sRGB color space. Before converting these photos to the XYZ color space, the individual R, G, and B values (ranging from 0 to 255) are normalized, limiting them to a narrower range of values (0 to 1). The use of a gamma function conversion equation is used to linearize the sRGB values, which then allows for their translation into linear RGB values. Using a transformation matrix (T), the linear RGB values are smoothly and accurately converted to the normalized XYZ values inside the XYZ color space. Converting the initial RGB data from the color camera to comparable XYZ values, and then estimating the PCA spectral components using the spectrometer, is crucial for improving the precision and consistency of our observations. Converting RGB to the XYZ color space yields a color representation that is not dependent on any specific device. This ensures that the spectral data remain consistent and can be compared accurately across many samples and experimental settings. Standardization is essential for attaining dependable outcomes in our analysis. It is crucial to emphasize that the sRGB color space often follows a specified white point, D65 (XCW, YCW, ZCW). This standard does not consider the characterization of the intrinsic white point (XSW, YSW, ZSW) of a light source. Therefore, it is crucial to apply a chromatic adaptation transformation matrix to the obtained XYZ data. This process adjusts the D65 white point to match the white point of the illuminating light source, resulting in a very accurate set of XYZ values when measured with the XYZ_Camera_. Normalization guarantees the constant scale of data across pictures, enabling consistent processing of pixel values. Gamma function conversion compensates for the non-linear correlation between pixel intensity and perceived brightness, hence improving the precision of intensity depiction. The process of converting to HSI allows for the separation of color information, namely hue and saturation from brightness or intensity. This separation enables more efficient manipulation of color and brightness as distinct entities. This transformation simplifies activities such as color analysis and segmentation by offering a more understandable depiction of color characteristics inside the picture. Collectively, these procedures enhance the interpretability and usefulness of HSI pictures in many image-processing applications.

Within the spectrometric domain, the transformation of reflection spectrum data into the XYZ color gamut space necessitates the establishment of the XYZ color matching function (CMF) in conjunction with the illumination source’s spectrum S(λ). Delineating that the Y component within the XYZ color gamut space directly corresponds to luminance is pertinent, engendering a direct proportionality. This scenario allows for the derivation of the luminance ratio, denoted as “k”, via Equation (1), which subsequently normalizes the luminance values to a standard of 100. The conversion of reflection spectrum data into the XYZ value (XYZ_Spectrum_) is achieved through the application of Equations (2)–(4). Upon establishing XYZ_Camera_ and XYZ_Spectrum_, a multivariate regression analysis is systematically executed to derive the requisite correction coefficient matrix (C) for camera calibration. The associated variable matrix (V) is subjected to analysis, including the discernment of parameters that can potentially contribute to camera-related discrepancies. These influential factors include, but are not limited to, nonlinearity inherent to the camera’s response characteristics, the presence of dark current within the camera, imprecise color separation that stems from the use of color filters, and the manifestation of color shifts during imaging
(1)$$k=100/\int_{380nm}^{780nm}S\left(\lambda \right)\overset{-}{y}(\lambda )d\lambda$$
(2)$$X=k\int_{380nm}^{780nm}S\left(\lambda \right)R\left(\lambda \right)x(\lambda )d\lambda$$
(3)$$Y=k\int_{380nm}^{780nm}S\left(\lambda \right)R\left(\lambda \right)y(\lambda )d\lambda$$
(4)$$Z=k\int_{380nm}^{780nm}S\left(\lambda \right)R\left(\lambda \right)z(\lambda )d\lambda$$

Once the camera calibration is finished, the XYZ_Correct_ values of the calibrated 24-color patch and the reflection spectrum data (R_Spectrum_) of the 24-color patch obtained by the spectrometer may be examined to obtain the transformation matrix (M). The R_Spectrum_ tool utilizes principal component analysis (PCA) to identify the primary principal components. RGB data offer a limited amount of color information across three wide channels, which is insufficient for conducting a thorough spectral analysis due to the lack of fine detail. By converting the data to the XYZ color space and utilizing PCA, we can isolate main components that capture the most important spectral characteristics. This methodology enables us to break down the spectral data into elements that precisely represent the fundamental fluctuations in the spectral profiles. This is essential for effectively modeling and forecasting the thickness of transparent glass sheets. Regression directly from RGB data to thickness levels would depend on the restricted information present in the RGB channels. By including PCA components produced from XYZ-transformed data, we may leverage more intricate spectral information, resulting in a more resilient and precise regression model. This technique mitigates noise and amplifies the signal related to variations in thickness, hence enhancing the forecast precision of our model. Subsequently, multiple regression analysis is conducted on the respective principal component scores and XYZ_Correct_ [33,34,35]. To reduce the dimensionality of XYZ_Correct_, the six most significant sets of primary components are used. These components are capable of accounting for 99.64% of the variance in the data. The associated main component score may be used for regression analysis using XYZ_Correct_. In the multivariate regression analysis of XYZ_Correct_ and score data, the variable V_Color_ is chosen based on its inclusion of all possible combinations of X, Y, Z, and M, and is derived using Equation (5). Furthermore, the utilization of the transformation and PCA procedure allows us to accurately synchronize our HSI data with measurements from a spectrometer of high precision. Ensuring the alignment is crucial in order to validate the quality of our spectral profiles and guarantee the reliable prediction of glass thickness using the spectral data in our model. The utilization of this approach is imperative in order to attain the elevated degree of accuracy needed for gauging the thickness of translucent glass sheets, as evidenced by the minimal root mean square error (RMSE) values and exceptional rates of precision in our findings. Next, XYZ_Correct_ data are input into Equation (6) to compute the simulated spectrum (S_Spectrum_). Lastly, S_Spectrum_ is contrasted with R_Spectrum_, and the mean square root error (RMSE) for each color patch is determined to be 0.63. The distinction between S_Spectrum_ and R_Spectrum_ may also be conveyed via variations in hue. The mean chromatic disparity is 0.75, and seeing the chromatic disparity is challenging. During the process of reproduction, colors are replicated with precision.
(5)$$\left[M\right]=\left[Score\right]\times pinv\left(\left[V_{Color}\right]\right)$$
(6)$${\left[S_{spectrum}\right]}_{380\sim 780nm}=\left[EV\right]\left[M\right]\left[V_{Color}\right]$$

The matrix [*EV*] in Equation (6) denotes the eigenvectors obtained from PCA, which are employed to convert the original data into the principal component space. This transformation does not represent an inverse PCA, but rather a linear projection onto the new axes determined by the principal components. For our study, we employed the initial six main components, which effectively captured the most substantial variation in the data. The VIS-HSI technology built through the preceding process can simulate the reflection spectrum from the RGB values captured by an industrial camera.

### 2.2. NIR Hyperspectral Algorithm (NIR-HSI)

The NIR-HSI used in the current study is created by merging data obtained from an industrial camera with NIR-HSI technology. This imaging technique works using electromagnetic waves with wavelengths ranging from 900 nm to 1000 nm, and it can distinguish between different wavelengths with a precision of 1 nm. NIR-HSI conversion allows for the integration of spectral data into images captured by an NIR camera. The investigation involves capturing images data using an industrial camera and obtaining the reflectance spectra of different material samples using halogen light. Following that, a computational framework is developed to build the NIR-HSI approach (see Appendix A for the schematic representation of the NIR-HSI algorithm). The industrial camera captures images combined with an LC filter, using its ability to detect and measure radiation within the wavelength range of 900 nm to 1100 nm. This allows it to acquire reflectance spectra in the NIR spectral domain. The reference targets used in this study include six different thickness values of glass that have been coated. The targets are exposed to radiation from a halogen lamp to obtain reflectance radiative data that are not affected by any external factors. Camera calibration is not necessary in this approach, since the sensor device is consistent over the whole spectral range. Once the reflectance spectra of the specimens are obtained, principal component analysis (PCA) is conducted on these spectra. The first six main components explain 99.79% of the variation in the data. The principal components are then used to infer the appropriate principal component scores. The scores are then used in the regression analysis in conjunction with the NIR spectral domain. A spectrum reconstruction transformation matrix is generated using the generalized inverse matrix approach to facilitate the alignment of digital values with spectral weights. This matrix converts the numerical values of an image into spectral distributions. This matrix allows for the immediate conversion of digital data from the camera into spectral profiles, without the need for labor-intensive spectral scanning at different points. Ultimately, the simulated spectra (S_Spectrum_) are compared to the observed reflectance spectra, allowing for the calculation of the RMSE for each target specimen. The calculated average RMSE is 0.001843. This research utilizes microscopic pictures to verify and align the simulated spectral profiles acquired from the analysis of HSI conversion. More precisely, the spectroscopic data obtained from microscopic imaging serve as a reference point for evaluating the precision of the simulated profiles created from the generalized inverse matrix approach applied to the normalized HSI pictures. In terms of equipment, the microscope does not have a color camera or a short-wave infrared camera with a filter. Alternatively, the microscopic images are recorded separately and subsequently contrasted with the spectral profiles acquired by the HSI technique. Employing a distinct methodology, it becomes possible to directly compare and verify the spectral data, guaranteeing the precision of the thickness measurements of the transparent glass sheets. Converting the images to spectral values before correlating them with the ground truth of thickness is crucial for improving accuracy and reliability. These changes accurately capture the intricate spectral features that are crucial for estimating the thickness of clear glass coatings. Conventional correlation approaches may overlook these nuances. By converting to spectral values, the data are standardized, guaranteeing uniformity across many trials and samples. PCA improves the extraction of features by emphasizing important spectral components, hence enhancing the efficacy of regression models in comparison to using raw photos. Furthermore, the alignment of HSI data with high-precision spectrometer measurements serves to verify the quality of the model, thereby guaranteeing dependable predictions of thickness. The VIS-HSI approach faces challenges when dealing with high transmittance in transparent glass coatings, which reduces the effectiveness of direct correlations. Converting to spectral values resolves this problem by highlighting important spectral characteristics, and guaranteeing accurate and dependable measurements of thickness for transparent coated glass and thin films using NIR-HSI.

### 2.3. Analysis Method for Glass Coating Thickness

The focus of this research is to examine a technique for identifying the thickness of glass coatings. The detection process entails obtaining images of glass that has been coated, using a microscope with a set resolution of 800 × 600 pixels. HSI analysis is used for each pixel in these images to include spectral information in the imagery. Afterwards, this spectral information is used along with the data obtained from a surface profilometer for regression analysis. The thickness of the glass coating is determined using the parameters obtained from the spectral information analysis. The signals are classified into seven distinct categories based on the thickness of the glass (namely unclassifiable noise signals that comprise the glass substrate, defects, and dust), transmittance signals for transparent thin films with thicknesses of 4000 nm, 5000 nm, 6000 nm, 7000 nm, 12,000 nm, and 14,000 nm.

## 3. Results

Optical microscopy (OM, (MM40, Nikon, LinTrading Co., Taipei, Taiwan)) was used in this experiment to capture images, which were then entered into the NIR-HSI analysis. Figure 3 illustrates the process of spectral modifications and corrections critical for accurately determining glass thickness. Each image produces unique spectral data, as seen in Figure 3a, illustrating the spectral modifications before correction. Since NIR-HSI data mostly represent spectra that are transmitted, it is necessary to remove the impact of the light source. Figure 3a presents the raw transmittance spectra of glass samples with varying thickness values (G4000, G5000, G6000, G7000, G12000, G14000) across the wavelength range of 800 nm to 1000 nm. These raw spectra show distinct peaks and variations in transmittance levels, which need correction to account for inherent spectral features and noise. This ensures that only the transmitted NIR spectral fingerprints are retained, as shown in Figure 3b. Figure 3b shows the spectral data after applying correction techniques to normalize and standardize the transmittance values. The normalized method is the spectrum of Figure 3a divided by the spectrum of glass without coating and containing the xenon lamp light source. This correction reduces noise and enhances clarity, resulting in more consistent and distinguishable features that correlate with glass thickness. The post-correction spectral curves demonstrate improved precision, making it easier to identify the specific characteristics associated with each thickness level. After being normalized, the corrected spectral information improves the clarity in differentiating the distinctive intensities included in the transmitted spectra. Through the examination of the post-corrected spectral data characteristics, each data point in every image, totaling 144,000 pixels, is analyzed. The method leads to the formation of scatter plots that show the variation in thickness for six various values of coated glass. This is seen in Figure 3c. These data confirm the capabilities of NIR-HSI technology to distinguish among samples in transparent thin films. Figure 3c displays the results of PCA applied to the corrected spectral data. PCA reduces the dimensionality of the data, extracting significant features and plotting them against the first two principal components. This visualization effectively clusters the data points based on glass thickness, showing clear separation between clusters corresponding to different thicknesses. This indicates that PCA has successfully captured the relevant spectral features for thickness estimation. The transformation from raw to corrected spectral data, followed by PCA, underscores the necessity of intermediate modifications. These steps enhance the accuracy and reliability of thickness measurements, demonstrating the importance of spectral corrections and feature extraction in our methodology. This process ensures precise and consistent data analysis, vital for the accurate determination of glass thickness.

This research aims to analyze spectra from thin glass sheets of different thicknesses using NIR-HSI technology. Figure 4 demonstrates the depiction of sample thickness by matching each spectral band with the appropriate changes in thickness, based on spectral disparities. Figure 4a,c,e,g,i,k show the OM images of the samples. Figure 4b,d,f,h,j,l display the visualization results of the samples, specifically showing the thickness of the images. The uncoated glass substrate is shown in a bright blue color on the left side of each image. In Figure 4b, the colors representing each spectral band (green) are used on the right side, showing noticeable differences in color for thicker and uneven coatings. Furthermore, a spectrum of orange shades becomes visible at the boundaries of the coatings due to variations in thickness. Figure 4b–l provide a consistent pattern, highlighting how the combination of NIR-HSI and visualization approaches simplifies and improves the identification of thin film thickness.

To validate the accuracy of the NIR-HSI method, images of glass samples were acquired from different areas, ensuring that they all had the same thickness. These images were then analyzed comparatively. In this investigation, a total of six glass samples with identical thicknesses were used, all of which were coated. Images of several locations were taken with a pixel resolution of 800 × 600, including a 4 cm^2^ area. For regional comparison, a total of 4.8 million data points were used, as seen in Figure 5a,b (using the G5000 sample as an example). The accuracy may be determined by conducting a regional study, as seen in Figure 5c. The findings indicate that the blue particles represent the pixel data in the G5000-1 image, whereas the red particles represent the pixel data in the G5000-2 image. The accuracy rate is determined by calculating the overlap of the red and blue spectral characteristics on both sides. Figure 5 displays the accuracy findings. The G4000 sample has an average accuracy of 81.3%. Similarly, the G5000 sample has an average accuracy of 82.7%. The G6000 sample has a mean accuracy of 76.5%, whilst the G7000 sample showcases an average accuracy of 80.06%. The G12000 sample has an average accuracy of 73.8%, whereas the G14000 sample exhibits an average accuracy of 75.1%. The mean accuracy rate for all the samples combined is 78.24%. As thickness grows, accuracy experiences a little drop. The decline in question may be ascribed to the reduction in the adjusted transmittance signal as the thickness increases, increasing judgment mistakes owing to the difficulty of handling reduced signal intensities. The failure to attain data accuracy rates of 90% might be ascribed to many variables. The presence of uneven edge thickness in the coated glass images may have compromised the accuracy of the evaluation procedure. Furthermore, the presence of intrinsic flaws in the glass substrate may lead to inaccuracies in the study. Furthermore, the lack of a controlled atmosphere, i.e., free from contaminants, may have caused the influence of dust or particle matter throughout the experiment.

## 4. Discussion

Optimized approaches are necessary to acquire reliable spectrum data due to the extreme transparency of glass. We modified the lighting conditions to reduce reflections and provide uniform illumination. Hyperspectral imaging (HSI) was employed to obtain precise spectral data spanning the visible (VIS) and near-infrared (NIR) bands. Principal component analysis (PCA) was applied to the hyperspectral imaging (HSI) data to identify and extract important spectral characteristics that can be used to differentiate differences in glass thickness. We verified the correctness of the gathered spectrum data by comparing them to measurements taken with a high-precision spectrometer. These methods guaranteed dependable data collection, precisely representing the variations in glass thickness, despite its high level of transparency. Although this study developed the VIS-HSI approach, the testing findings were disappointing because the transparency of the glass impeded precise detection. Nevertheless, this first progress was essential as it established the foundation for the NIR-HSI approach, which is founded on comparable concepts. The NIR-HSI method effectively addressed the shortcomings of VIS-HSI and produced highly encouraging outcomes. Hence, although the experimental section mainly showcases NIR-HSI outcomes, the VIS-HSI approach is elaborated upon to offer background information and emphasize the iterative process that resulted in the successful NIR-HSI technique. For future studies, it is recommended to obtain specimens of transparent coated glass with even smaller thickness dimensions while considering the samples. Developing an advanced NIR-HSI approach using more accurate data may result in enhanced and nuanced detection capabilities. The hardware used in the present research is not suitable for successful integration into the glass industry’s manufacturing processes due to the tiny size of the measured area. Hence, it is essential to enhance the hardware using industrial cameras with the ability to capture wider image regions. This improvement will enable the scanning of larger areas with more flexibility, increasing the present scanning area of 4 cm² to over 20 cm². This will effectively satisfy the requirements of product inspection in the glass sector. This experiment demonstrates a clear pattern of diminishing accuracy as the thickness of the specimens increases. The decrease in NIR light transmission is responsible for the weakened signals of acquired spectral data, leading to this drop. When choosing the light source spectrum, it is preferable to use data on transmittance from different spectral bands to improve the accuracy of signal reception. Hence, our focus will be on enhancing the precision of thin film detection to achieve a level of accuracy of 95%. While this study focuses on a specific imaging area served to establish proof of concept, acknowledging the necessity of scalability in real-life applications is crucial. Expanding the imaging area would ensure the technique’s applicability across broader industrial scenarios. Additionally, the highlighted challenge of low light intensity underscores the need to address practical complications for reliable implementation. Strategies to mitigate these challenges, such as optimizing camera settings or incorporating supplemental lighting sources, warrant further exploration. Future research directions should aim to bridge the gap between laboratory experimentation and real-world industrial settings, incorporating feedback from stakeholders to refine and optimize the proposed HSI technique for practical deployment. While using data on transmittance from different spectral bands could enhance accuracy, it also prompts the question of whether a wider spectrum should be employed. In this study, the use of an NIR monochromatic camera to simulate NIR spectra over a larger wavelength range may not fully capture the complexity of the spectrum. To address this, additional methods could be explored to obtain a more comprehensive spectrum, such as using multiple light sources with varying wavelengths or incorporating advanced spectral imaging techniques. These approaches would enable more accurate characterization of materials and enhance the performance of HSI systems in industrial applications. Employing an HSI camera for the examination of glass films has the potential to yield more comprehensive and precise spectral data. Nevertheless, our method of converting RGB data to XYZ and implementing PCA presents numerous benefits. HSI cameras are costly and have limited availability compared to color cameras. Our approach utilizes easily obtainable industrial cameras, resulting in a more economical and attainable solution. The conversion to XYZ guarantees a device-agnostic and standardized depiction of color data, ensuring uniformity across various circumstances and specimens. By performing PCA on the XYZ-transformed data, we are able to identify and utilize important spectral characteristics that enhance the reliability and precision of our regression model, as evidenced by the low RMSE values and high accuracy rates we have achieved. Furthermore, our method ensures that HSI data are in agreement with precise spectrometer measurements, thus confirming the correctness of our spectral profiles. This technique enables accurate measurements of the thickness of clear glass sheets without requiring specialized or costly equipment.

## 5. Conclusions

We presented a technique for measuring six different thickness values of transparent coated glass using the VIS-HSI and NIR-HSI techniques. Through the simulation of the VIS results, the simulated spectral images exhibit an average color difference of 75%, indicating a high level of accuracy, employing PCA to extract feature vectors and subsequently calculate eigenvalues to obtain transformation matrices in the construction of the VIS-HSI technique. However, when attempting to measure the thickness of transparent glass films, the average transmittance of visible light is 88%, causing the transmittance to be excessively high and hindering the effective separation of feature signals for identification. Consequently, the VIS-HSI technique cannot be utilized for distinguishing transparent glass film thickness. In the establishment of NIR-HSI, PCA was used to extract feature vectors, followed by calculating eigenvalues and obtaining transformation matrices. The simulated spectral images and the measured spectral information exhibit an average RMSE of 0.1843%, indicating a high level of accuracy. Thus, the transmittance feature signals in the NIR spectrum can be extracted utilizing the NIR-HSI technique to distinguish Low-E transparent glass film thickness, achieving an accuracy rate of up to 85%. However, the weakening of transmittance signals with increasing thickness must be addressed to overcome the issue of low signal intensity and to enhance accuracy. In addition, the utilization of image thickness visualization techniques allows for a clear presentation of detection results, achieving accurate outcomes in large-area experiments. The outcomes of this experiment are envisioned to provide a new approach for the automated optical inspection industry of coated glasses. Complete automation and large-area detection can be achieved through NIR-HSI and image-based thickness recognition techniques, transforming from traditional point and line scanning methods to the 4 cm² area scanning method utilized in this experiment. This technique not only saves on labor and time costs but also significantly enhances production capacity while ensuring product quality and increasing industrial value.

## Figures and Tables

**Figure 1 sensors-24-05094-f001:**
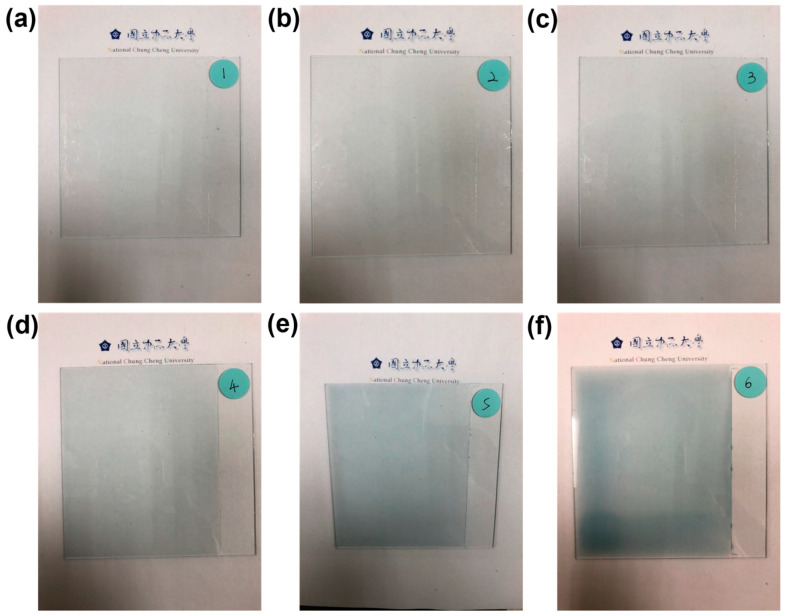
Experimental samples (**a**) G4000; (**b**) G5000; (**c**) G6000; (**d**) G7000; (**e**) G12000; (**f**) G14000.

**Figure 2 sensors-24-05094-f002:**
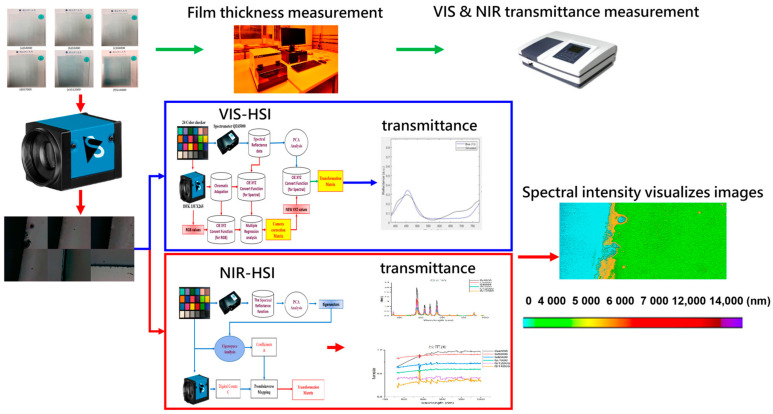
Flowchart of the rapid large-area film thickness identification.

**Figure 3 sensors-24-05094-f003:**
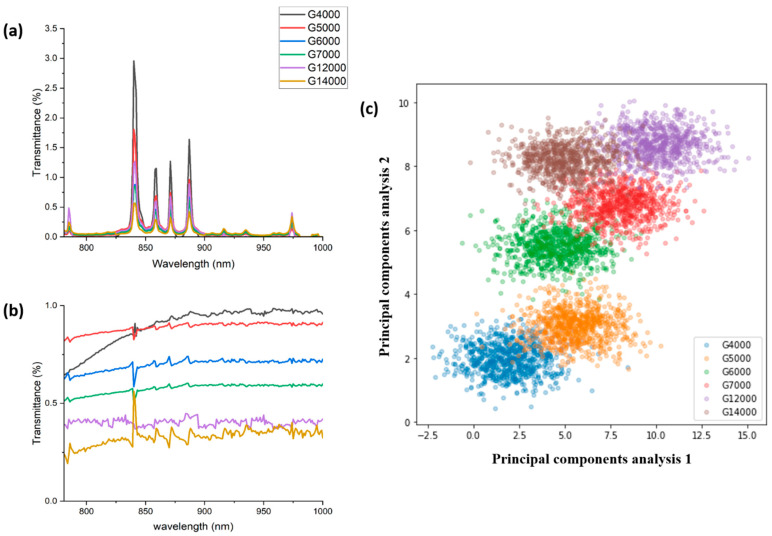
(**a**) Transmittance in % of different types of samples. (**b**) Transmittance of different samples after eliminating all noise. (**c**) Scatter diagram of the spectral regions of the six experimental samples.

**Figure 4 sensors-24-05094-f004:**
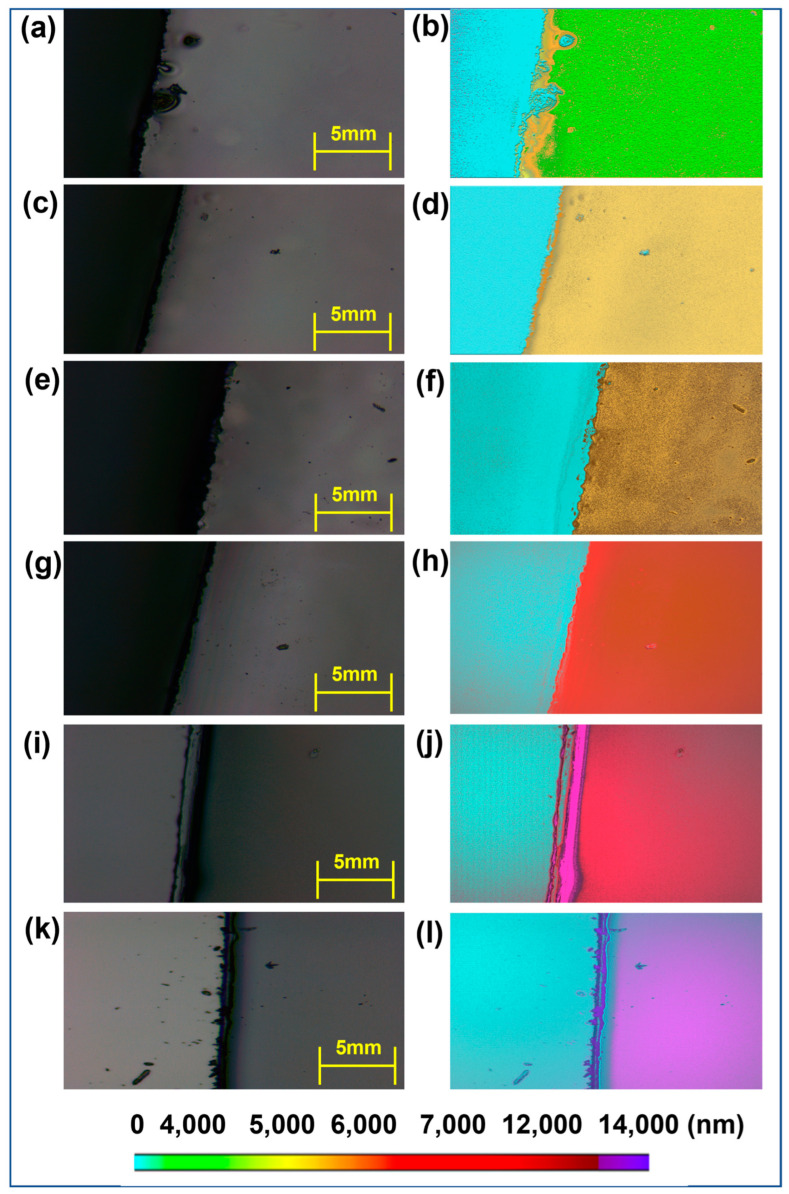
OM images of the samples (**a**) G4000, (**c**) G5000, (**e**) G6000, (**g**) G7000, (**i**) G12000, and (**k**) G14000. Image thickness visualization results of the samples (**b**) G4000, (**d**) G5000, (**f**) G6000, (**h**) G7000, (**j**) G12000, and (**l**) G14000.

**Figure 5 sensors-24-05094-f005:**
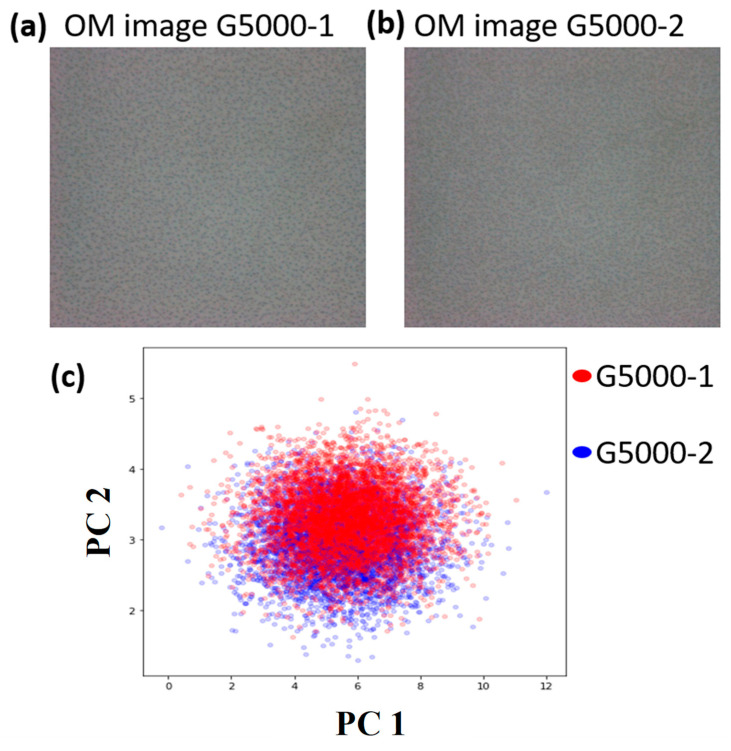
(**a**) OM image of G5000-1; (**b**) OM image of G5000-2; (**c**) overlapping red and blue spectral characteristics of two samples.

## Data Availability

The data presented in this study are available upon request to the corresponding author.

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
