# Peer review of "Large-Area Film Thickness Identification of Transparent Glass by Hyperspectral Imaging"

_sensors, 2024, doi:10.3390/s24165094_

Round 1

Reviewer 1 Report

Comments and Suggestions for Authors

The study proposes the use of a color camera or broadband NIR to estimate the values of glass thickness as measured by the surface profilometer and known thicknesses.

The use of the Beer-Lambert equation on the simulated spectral profiles is mentioned in the introduction, however this is nowhere explained neither in the paper nor in the supplementary material.

It is also not well explained how the microscopic images are used, and whether the microscope is combined with the color camera or the SWIR camera with a filter (this does not seem to be the case).

Overall, it is not clear to me what is the added value of transforming the initial RGB information from the color camera to equivalent XYZ, and then to transform/estimate the PCA spectral components measured with the spectrometer, since no additional information can be created that was not present in the original color data, only a model transformation is achieved. The regression model towards the measured thickness levels could be done directly from the original color data (whether in RGB, XYZ or Lab format) invalidating the need for this approach.

Alternatively, a truly hyperspectral camera could be used to inspect the glass films, if it was expected to better estimate the thicknesses than the color camera.

Figure 3a is supposed to illustrate spectral modifications and correction, but this is not clearly seen from the Figure neither explained.

The authors conclude that transparent coated glass thickness can be estimated with VIS-HSI but this is not possible with transparent glass films. With NIR-HIS (estimating NIR spectrum from a broadband NIR image) seems to be possible for low-thickness films. However, it is not clear why they intermediate transformations to spectral values are required instead of a direct correlation between the either color/broadband images and the thickness ground-truth.

Comments on the Quality of English Language

It could sometimes be a bit more concise English, to be more understandable and suitable for scientific publications.

Reviewer 2 Report

Comments and Suggestions for Authors

After reading this paper, the author made an exploratory study on the thickness of the glass film. This study looks interesting, but there are some questions that need to be discussed. There are mainly the following problems, but also more serious problems:

1. The title and content of this article both mention the HSI method, but from what I read, it seems to be a separate combination of camera and spectrometer. I am puzzled about how the data acquisition system used in this paper collects data;

2. For glass, due to its high transparency, it is difficult to collect spectral data, and the author needs to explain the collected data in more detail;

3. When conducting thickness analysis, how to determine the relevant thickness according to the data, what algorithm was used, and how the process was, were not clearly explained.

Comments on the Quality of English Language

The author needs to give a detailed description of the system and related algorithms used in the paper, and the English expression can be improved.

Reviewer 3 Report

Comments and Suggestions for Authors

In this manuscript, the authors present a novel method for detecting and measuring transparent glass sheets through the hyperspectral imaging (HSI) technique. The authors used this method to create two hyperspectral algorithms: the Visible Hyperspectral Imaging Algorithm (VIS-HSI) and the NIR Hyperspectral Imaging Algorithm (NIR-HSI). They then did a series of tests to prove that the NIR-HSI method worked. The manuscript description is not detailed enough, and the content arrangement is not reasonable. Therefore, I think that the current manuscript is not appropriate for publication in this journal. Some issues should be addressed:

  1. There are some typing or display errors in the manuscript; for instance, in Page 8 Line 298, "cm2".
  2. Most of the figures in the manuscript are blurry, especially Figure 2.
  3. Figure 2's flowchart for NIR-HSI differs from the supplementary material in Figure S2 and the description in the manuscript.
  4. The manuscript provides a detailed description of VIS-HSI, but the experiment section solely displays the NIR-HIS results, while the conclusion section only briefly mentions the VIS-HSI results. Supplementing the details of the VIS-HSI experiment is necessary.
  5. In the manuscript’s Equation (6) and supplementary material’s Equation (S16), what does the matrix [EV] do? Is it the inverse transformation of PCA or some other operation?
  6. I believe there should be more photos of the experimental devices.
  7. In the experiment, the authors reconstructed the HSI data using images captured by an RGB camera and a SWIR camera, and you measure the true transmission spectrum of the samples with spectrometers. However, the manuscript solely displays the RMES difference between the reconstructed and true values, and I believe it's necessary to include a direct comparison between the reconstructed and true values.
Comments on the Quality of English Language

No

Round 2

Reviewer 1 Report

Comments and Suggestions for Authors

I appreciate the effort done to provide more explanations and it is a bit clearer their goal in the paper and the method used. The fact that for XYZ conversion, simulating the spectral range of 380-780 is required, and this to have a color measurement that is device independent. However, it is the HSI-NIR method, measuring in the 800-1000 nm range that correlates better with the thickness.

As for the references, it is still not explained what the drawbacks are of the color-based systems in literature and references [30] to [32] are self-references about medical applications, quite unrelated to the application in this paper, so that could be improved.

I agree that going to the XYZ space (for which the help of the spectrometer is needed for the spectral estimation) is more robust and device independent than RGB. Then, PCA components can be extracted from the estimated spectra, which help visualize the spectral clustering of the different thicknesses. It remains a bit unusual to go from a three-dimensional space of XYZ to 6 PCA components, while typically PCA is used to reduce dimensionality. But I suppose that expressing XYZ in a different vectorial base of combinations of XYZ can help visualize the clustering for different thicknesses.

 I agree as well that if a model can be achieved in this approach with a color camera, and the initial help of a spectrometer, then it can be more cost-efficient than the use of a hyperspectral camera.

With respect to the comment on Figure 3, I am afraid it is still not well addressed. My point was that no explanation is given on the specific kind of correction or normalization method applied.

 About the microscope used, I understand it is not couple with a hyperspectral camera, but it is mentioned that “the spectroscopic data from microscopic imaging provides a benchmark for assessing the accuracy of the simulated profiles”. Therefore, I understand they are using for instance Fourier transform imaging spectroscopy, since in some way the microscope can give a spectral measurement? This is not explained clearly in the paper and no reference to the optical microscope model is given.

The authors claim that “Direct regression from RGB data to thickness levels would depend on the restricted information present in the RGB channels.” Still, it would have been interesting and stronger argument to see the comparison and the real gain in the thickness estimation from the spectral estimation, since the RGB data is already available.

Comments on the Quality of English Language

As it was remarked in the first review, their English use and sentence building could be more concise and clearer. It does not read as clear as many other papers do.

Reviewer 2 Report

Comments and Suggestions for Authors

The authors made enough efforts to represent this study, suggestions gived follows:

1.Regression model was applied in this paper, but no clear result gived in the modified manuscript, only one sentence in abstract showed to the readers. Please provide  one result table can be very well.  

The root-mean-square error 29 inside the NIR area is impressively low, measuring just 0.02.

2. The answer of question 3 is not suitable. I'd like to know how you conduct your experiments in detail, not what methonds you used.

Comments on the Quality of English Language

 Minor editing of English language required

Reviewer 3 Report

Comments and Suggestions for Authors

All the issues I concered have been addressed. I think it can be published in the journal.
